# Managing Difficulties of Microsatellite Instability Testing in Endometrial Cancer-Limitations and Advantages of Four Different PCR-Based Approaches

**DOI:** 10.3390/cancers13061268

**Published:** 2021-03-12

**Authors:** Janna Siemanowski, Birgid Schömig-Markiefka, Theresa Buhl, Anja Haak, Udo Siebolts, Wolfgang Dietmaier, Norbert Arens, Nina Pauly, Beyhan Ataseven, Reinhard Büttner, Sabine Merkelbach-Bruse

**Affiliations:** 1Institute of Pathology, University Hospital Cologne, D-50924 Cologne, Germany; birgid.schoemig-markiefka@uk-koeln.de (B.S.-M.); theresa.buhl@uk-koeln.de (T.B.); reinhard.buettner@uk-koeln.de (R.B.); sabine.merkelbach-bruse@uk-koeln.de (S.M.-B.); 2Institute of Pathology, University Hospital Halle (Saale), D-06112 Halle, Germany; anja.haak@uk-halle.de (A.H.); udo.siebolts@uk-halle.de (U.S.); 3Institute of Pathology, University Regensburg, D-93053 Regensburg, Germany; wolfgang.dietmaier@ukr.de; 4Center for Histology, Cytology and Molecular Diagnostics Trier, D-54296 Trier, Germany; arens@molekularpatho-trier.de; 5Department of Gynecology and Gynecologic Oncology, Evang. Kliniken Essen-Mitte, D-45136 Essen, Germany; n.pauly@kem-med.com (N.P.); b.ataseven@kem-med.com (B.A.); 6Department of Obstetrics and Gynecology, University Hospital, LMU Munich, D-81377 Munich, Germany

**Keywords:** microsatellite instability, mismatch repair deficiency, polymerase chain reaction, immunohistochemistry, endometrial cancer

## Abstract

**Simple Summary:**

Due to the approval of immune checkpoint inhibitor therapy for microsatellite instability-high or mismatch repair-deficient advanced solid tumors, testing of both biomarkers has gained interest in recent years. Available testing systems were established in the context of Lynch Syndrome for colorectal cancer, thus differences between microsatellite profiles across cancer types may lead to false data interpretation using validated tests for another tumor entity. The present study deals with challenges during microsatellite instability testing in endometrial cancer (EC) and provides a comprehensive comparative study of four different PCR-based approaches which could help to improve microsatellite instability (MSI) testing in future screenings. A validation strategy has been developed for the Idylla system, which can guide the method transfer to other tumor entities, and a screening procedure for EC has been proposed. By direct comparison, this study was able to highlight advantages and limitations of each system in an extensive manner.

**Abstract:**

Microsatellite instability (MSI), a common alteration in endometrial cancers (EC) is known as a biomarker for immune checkpoint therapy response alongside screening for Lynch Syndrome (LS). However, former studies described challenging MSI profiles in EC hindering analysis by using MSI testing methods intensively validated for colorectal cancer (CRC) only. In order to reduce false negatives, this study examined four different PCR-based approaches for MSI testing using 25 EC samples already tested for mismatch repair deficiency (dMMR). In a follow up validation set of 75 EC samples previously tested both for MMR and MSI, the efficiency of a seven-marker system corresponding to the Idylla system was further analyzed. Both Bethesda and Promega marker panels require trained operators to overcome interpretation complexities caused by either hardly visible additional peaks of one and two nucleotides, or small shifts in microsatellite repeat length. Using parallel sequencing adjustment of bioinformatics is needed. Applying the Idylla MSI assay, an evaluation of input material is more crucial for reliable results and is indispensable. Following MMR deficiency testing as a first-line screening procedure, additional testing with a PCR-based method is necessary if inconclusive staining of immunohistochemistry (IHC) must be clarified.

## 1. Introduction

Endometrial cancer (EC) is one of the most common gynecologic malignancies worldwide [1]. To date, there have been several studies of molecular subtyping with the intention to offer detailed prognostic information. The most elaborate version is provided by the Cancer Genome Atlas, dividing ECs into DNA polymerase epsilon (POLE) mutated, copy-number low, copy-number high and microsatellite instability (MSI) tumors [2]. The latter is caused by aberrations in genes encoding mismatch repair (MMR) proteins, including MutL homolog 1 (MLH1), MutS protein homolog 2 (MSH2), MutS protein homolog 6 (MSH6) and PMS1 protein homolog 2 (PMS2) [3,4,5,6]. Following mutation, an uncontrolled accumulation of mutations within so-called microsatellites—repetitive sequences distributed throughout the genome—can sustain tumor formation [7]. The MSI phenotype occurs in up to 30% of EC and can be caused by sporadically or inherited occurring mutations in the MMR system [8]. The inherited autosomal dominant disorder, defined as Lynch Syndrome (LS), is associated with higher lifetime cancer risk [9] and constitutes around 3% of EC [10]. The prognostic impact of inherited or sporadic EC caused by MSI is still contradictorily discussed [11,12]. Analysis regarding chemotherapy sensitivity are still ongoing [13]. MSI was recently designated as a predictive biomarker for to immune checkpoint therapy response [14,15]. Thus, clinical relevance of MSI and mismatch repair deficiency (dMMR) testing increased. Most of the testing methods initially developed for CRC are nowadays used for other tumor entities to an increasing extent [16]. Those methods comprise IHC for MMR proteins analysis [17] and PCR-based methods for MSI detection, separating tumors into MSI-high (MSI-H), MSI-low (MSI-L) and microsatellite stable (MSS) [18]. PCR-based detection approaches include the Bethesda panel, already agreed on several conferences [19], as well as commercial panels [20]. In recent years, next generation sequencing (NGS) panels [21] as well as fully automated systems have been evaluated [22]. Although using partially different marker systems, they all were initially validated for CRC. Since general screening of MSI and dMMR is also recommended for EC [16], a discussion about the best suited screening tool has started [23]. While high concordance is found between molecular and IHC analysis in both entities, the value decreases from 98% for CRC to around 94% for EC [24]. Consequently, verified screening tools for one tumor entity cannot be adopted without precaution regarding abnormalities for another. Troublesome interpretations of IHC caused by aberrant patterns, variations in intensity and heterogeneous loss of one marker due to, for example, intratumor heterogeneity are known for both CRC and EC [25,26]. In contrast, using PCR-based screening tools for MSI testing appears to be more complicated in EC [27,28].

Former investigations already mentioned a lower proportion of unstable markers for ECs and, in MSI-L cases, germline mutations could not be excluded completely in contrast to most CRC MSI-L samples [29]. Another crucial distinction between EC and CRC is a discrepancy in their MSI profiles. In EC, the phenomenon of smaller size deletions or insertions in microsatellite regions resulting in a smaller number of additional peaks visualized via fragment length analysis was already described [27] and brought into focus again due to the efforts for harmonization of MSI testing methods in EC. A reduced sensitivity for MSI detection by PCR could be traced back to the peculiarity that alterations of one nucleotide (nt) were missed during analysis [28].

Hence, aiming at avoiding false negatives to guarantee patient’s best treatment options in the future, appropriate MSI testing of EC samples is a hallmark of good clinical and laboratory practice. The aim of the present study was to assess the performance of the Idylla system and Next generation sequencing (NGS) with a custom GeneRead V2 panel (Qiagen, Hilden, Germany) in analyzing EC with known MMR status in comparison to the putative reference methods of IHC, an in-house Bethesda panel or the MSI Analysis System, Version 1.2 from Promega. By that, pitfalls during examination were uncovered and all methods were checked regarding applicability in routine diagnostics. To overcome consequences of smaller size deletions and insertions during NGS analysis, average numbers of deleted bases of 21 CRCs were directly compared to those of EC and a new threshold range was proposed. Since discrepancies compared to the other MSI systems occurred when using tumor slices for the Idylla system, a larger cohort of 75 additional EC samples was analyzed for validation purpose. Thereby, limitations were pointed out in order to simplify a solid integration of the system.

## 2. Results

### 2.1. Cohort One

#### 2.1.1. Immunohistochemistry

Four samples (Table 1: sample 12, 19, 20 and 23) showed a loss of expression only in PMS2. Subsequent analysis exhibited that three out of four samples were MSS. However, sample 12 was indeed MSI-H. Additionally, sample 18 with low intensity of PMS2 showed an MSI-L status in three out of four PCR analysis systems after reanalysis. Sample 17 with MSH2 loss and partial loss of MSH6 turned out to be MSI-H after re-evaluation of three PCR methods. The sole loss of MSH6 in sample 14 revealed an MSI-H status by PCR. Partial loss of MSH2 (sample 21) and low intensity of MSH2 and MSH6 staining (sample 22) ended up in an MSS status. Weak expression of PMS2 and MLH1 (sample 13) turned out to be ended MSI-H. Samples designated as being clearly dMMR or pMMR by IHC were all confirmed by all four PCR testing methods.

#### 2.1.2. In-House Bethesda Panel

Concordance between the in-house Bethesda panel and MSI status ascertained by the re-evaluated results of three additional testing methods (Promega, Idylla, NGS) amounted to 92%. Re-evaluation of all samples revealed a misclassification of two samples (samples 2 and 17) with deviating MSI status. Therefore, concordance level was adjusted to 100%. Additionally, subsequent analysis of sample 12 showed instability of one additional marker which was missed in the initial evaluation. A possible complex MSI profile in EC leading to misinterpretation indicated by only a small left shift is exemplarily shown for sample 1 in Figure 1.

#### 2.1.3. MSI Analysis System, Version 1.2 (Promega)

Comparison between the Promega panel with the results of the Bethesda panel and Idylla after re-evaluation yielded in concordance of 80%. Discordant results were clarified by a second molecular biologist after blinded re-evaluation. Samples 10, 15, 16, and 17 were corrected from MSS to MSI-H status and sample 18 from MSS to MSI-L, respectively. Thus, re-examination ensured a concordance level of 100%.

The differences between an obvious MSI-H status indicated by up to -8nt additional peaks (sample 4, Figure 2B), and only -1nt to -2nt more peaks detected in sample 10 (Figure 2C), are shown in Figure 2.

#### 2.1.4. Idylla Slice and DNA

At first overall concordance of the Idylla system was 88%. Samples 10, 15 and 16 were tested again using 200 ng DNA each, elevating concordance to 100%.

#### 2.1.5. GeneRead V2 Panel

NGS resulted in a concordance of 80% if using the threshold range known for CRC. In detail, two out of 25 samples showed a false negative result represented by a factor of 2.6. MSI status of samples 10, 15 and 17 was uncertain. One out of 25 samples could not be evaluated at all due to low coverage (below 200). The average of the deleted bases of MSI-H samples seemed to be lower than yet expected from CRC samples. To examine this observation, average factors of deleted bases of 21 unbiased CRC tumor samples were directly compared to results of EC samples analyzed with NGS of both cohorts within this study (Table 2).

Indeed, statistical analysis revealed that the average of deleted bases is significantly lower (*p*: 0.0001) in EC than in CRC (Figure 3). After adjustment of a new classification range, the overall concordance was increased to 100%. In detail, in contrast to CRC, MSI-H is represented by a factor of ≥2.6 and MSS with a factor of up to 2.4 in EC.

In summary, all tested MMR and MSI detection systems showed high overall concordance after reanalysis. Uncertain MMR status could be solved by PCR. For cohort one all deviating results for MSI using the Idylla system were resolved by increasing the amount of input DNA. Since average of deleted bases in EC samples were lower than in CRC, false negative or inconclusive NGS results were clarified after re-adjustment of factor classification.

### 2.2. Cohort Two

#### 2.2.1. Immunohistochemistry

The sole loss of expression in PMS2 in sample 55 shown in Table 3 turned out to be MSS. Samples 45 and 49 without expression solely in MLH1 were both MSH-H. MSI-H was also proven for sample 48, with loss of expression only in MSH6. Sample 57 with a weak expression pattern of all tested mismatch proteins resulted in MSS. Partial loss of MLH1 and PMS2 in sample 52 revealed MSI-H. Surprisingly, sample 23 with loss of expression in MLH1 and PMS2 turned out to be MSS with both testing systems, but showed a DNA methylation of the MLH1 gene promoter in subsequent analysis (data not shown).

#### 2.2.2. Idylla Slice and DNA

Using slices of 10 µm with a tumor area of 25–100 mm^2^ resulted in an overall concordance of 88% with the Idylla system compared to the other test systems. Three out of nine discordant samples were estimated as being MSI-L due to a sole instability in DIDO (sample 6), ACVR2A (sample 17) or MRE11 (sample 19). In order to elucidate the discrepancy, samples 22 and 37 were tested again using 200 ng DNA each. After reanalysis a mutation in MRE11 could be estimated for sample 22. Sample 37 remained to be MSS with the Idylla system. Unfortunately, four samples (samples 2, 38, 45 and 50, Table 2) could not be tested again with the Idylla system with extracted DNA directly, since the required amount of 200 ng DNA was not achieved (sample 2 = 1.46 ng/µL; sample 38 = 0.1 ng/µL; sample 45 = 0.3 ng/µL and sample 50 = 0.11 ng/µL). MSI-H status of samples 2, 22, 38, 45 and 50 was validated via NGS (Table 2). Sample 37 was depleted after re-analysis with extracted DNA using Idylla.

## 3. Discussion

In the last years, not only screening for Lynch Syndrome obtained primary focus, but also estimating MSI as a biomarker for immune checkpoint therapy has become more and more essential in EC. Thus, identification of reliable testing strategies is crucial for the best patient treatment options. Currently, IHC is widely recommended as a robust initial testing tool for the detection of dMMR in EC [30]. However, IHC still has its limitation due to inconclusive staining results in some of the samples. Using PCR-based technologies; MSI status of all 28 samples with uncertain MMR status could be clarified. Nevertheless, our results showed high concordance between dMMR being MSI-H (52 out of 53) as well as pMMR being MSS (19 out of 19). This indicates that IHC is indeed a suitable method for a fast and cost efficient first stage screening tool.

While PCR is at least as sensitive as IHC, methods are hampered by either complex data interpretation or constitutive validation in EC. In the first part of our study we compared four different PCR-based methods to point out technical limitations and immanent difficulties in analyzing EC. Additionally, a broader cohort of EC was used as validation cohort for the Idylla test system that we introduced into routine diagnostics procedures.

A drawback of using both Bethesda and the Promega panel initially established for CRC became evident due to more complex MSI profiles. Complex MSI profiles of EC reasoned by either small additional peaks of only 1 or 2 nt (Figure 2), or a small overall shift, as demonstrated in Figure 3, led to false data interpretation and demonstrated that knowledge of different MSI profiles in different tumor entities plays a key role for good laboratory practice [28]. Accumulations of smaller deletions and insertions in EC is supposed to originate in different timings of tumor development [31]. This assumption was not further investigated within this study due to limited cohort size but should be further analyzed in a larger cohort of EC.

The numbers of informative MSI results of the NGS panel was lower as compared to the remaining test systems (Table 1) if using the threshold region for CRC. This might be due to a significantly decreased number of deleted bases in microsatellite regions evaluated in an explorative manner of 21 EC in comparison to 21 CRC within this study. By adjustment of a new threshold range, concordance between NGS and other methods were increased to 100%. Nevertheless, a more precise investigation of a larger cohort is certainly needed. Alternatively, microsatellite markers may be added to the panel to get a more comprehensive overview of different loci [32]. Additionally, a broader spectrum of microsatellite loci could simplify future analysis [16,33].

Discordant results found with Idylla were more associated with testing preconditions. A critical factor is the sensitivity of the system depending on applied tumor tissue and cellularity. Following manufacturer’s instructions using at least 25 mm^2^ of a 10 µm tissue slice with at least 20% tumor cell content turned out to be not enough for all EC samples. Results of three samples (Table 1) of cohort one and one sample of cohort two (Table 3) were adjusted after applying 200 ng of extracted DNA.

Former investigations already mentioned the possibility of false negatives in MSI PCR-based testing approaches if using less than 30% tumor cellularity [28]. Therefore, we would recommend a higher amount of tumor tissue (50 mm^2^, 10 µm slice) with approximal 40% tumor cell content.

Four out of 42 samples (Table 3) classified as MSI-H/dMMR turned out to have only one instable marker if using Idylla. According to the manufacturer’s classification system and its biological relevance postulated for CRC, MSI-L is comparable to MSS and correspondingly assigned [34]. Though, we could show that instability of one marker could indeed indicate MSI-H in EC if using the Idylla system. The training of the initial integrated neuronal network of Idylla was done mostly with CRC samples. Therefore, an accumulation of small shift events in EC could possibly feature lower MSI scores. Thus, in contrast to CRC, MSI-L in EC detected via Idylla should be reviewed with an additional PCR or IHC for validation purpose in future. Recently, another study demonstrated a 100% sensitivity and specificity of Idylla MSI testing in EC using MSI-H samples with at least 40% tumor cell content [35]. However, using more comprehensive cohorts of 68 MSI-H and 32 MSS samples, we were able to confirm the specificity of 100% but revealed a sensitivity of 92.65% after reanalysis and inclusion of MSI-L as being MSI-H. Additionally, we could show that samples with poor DNA quality or low tumor cellularity could lead to false negatives in MSI status (Table 3).

Besides standardized data evaluation and a workflow without the need of paired normal tissue, a major advantage of the Idylla system is its short turnaround time. Both the Bethesda and Promega panel need at least three and NGS requires approximately seven working days of lab work and data interpretation. Consequently, keeping in mind the lower sensitivity, Idylla could be an easy-to-handle fast track opportunity in MSI testing. Especially for smaller laboratories without space for a large amount of technical equipment and for those hampered by time management due to limited number of staff, this system could be a suitable fast track MSI testing tool.

The enhancement of alternative treatment options requires a rapid and sensitive detection of dMMR and/or MSI-H in EC. Comparing pros and cons of testing systems commonly used in routine diagnostics, we would like to propose a screening strategy for EC as follows: Since IHC is a sensitive and still the most cost-effective method to detect dMMR, initial analysis should be performed in all EC patients. Already described for CRC and also shown within this study, the loss of expression in only one marker could be related to either MSS or MSI-H in EC [36]. For that reason, follow up testing with a PCR-based method is recommended. Additionally, blurred staining results must be verified using PCR. Idylla provides fast and reliable results if an overall tumor area of more than 50 mm^2^ is available and the tumor cell content is higher than 40%. MSI-L samples should be tested with an alternative method favorably using the Bethesda or Promega panel if paired normal tissue is available.

## 4. Material and Methods

### 4.1. Study Design

Within this study, two cohorts of a total of 100 ECs upfront diagnosed immunohistochemically as dMMR, mismatch repair proficiency (pMMR) and with uncertain status were analyzed. Cohort one (Table 1) comprised 11 ECs with dMMR, one with pMMR, 13 with uncertain MMR status. This cohort was enriched for challenging cases in order to reveal potential complexities of the systems. The MSI status of all samples in cohort one was determined by four different MSI assays. Subsequently, all MSI profiles tested with the in-house Bethesda and the Promega panels were evaluated independently by different molecular biologists. The re-evaluation was blinded. As the threshold of the average number of deleted bases is crucial for NGS, two cohorts of MSI-H CRC and EC were compared. For the validation cohort two (Table 3) consecutive samples from routine diagnostics were analyzed comprising additional 42 ECs with dMMR, 18 ECs with pMMR and 15 samples with uncertain MMR status as explained above. MSI status was detected by either the in-house Bethesda or the Promega panel and juxtaposed with the results of the Idylla system. Samples with known pMMR status were not tested with an additional PCR method excluding samples 58, 62–72, 74 and 75. Discordant results of both cohorts using the Idylla system were re-evaluated using extracted DNA of selected samples if an amount of 200 ng was accessible. Except of sample 37, all discordant samples were additionally analyzed via NGS in order to confirm the results of MSI-PCR by a second PCR method. For sample eight and nine of cohort one (Table 1) no slices of EC tumor sample were available, thus DNA extracts were tested directly.

### 4.2. Sources of Specimens

For the comparison of four different PCR-based testing methods 25 ECs immunohistochemically diagnosed as dMMR, MMR-proficient (pMMR) and with uncertain status were selected. For efficiency testing of the Idylla method, an additional 75 endometrial tumors diagnosed as dMMR as well as pMMR, and with uncertain status, were assorted. MSI status was determined by PCR and fragment length analysis using the Bethesda or the Promega panel. To determine differences between the average of deleted bases in CRC and EC, 21 samples of each entity showing MSI-H were compared (Table 2). An overview of the clinical-demographical characteristics from all patients included within this study is shown in Appendix A. The study protocol conformed to the ethical guidelines of the 1975 Declaration of Helsinki, as reflected by the approval of the institution’s human research review committee (Ethics Committee of the Medical Faculty of University of Cologne: registration no. 13-091). Patients gave their written consent to usage of their tumor specimen.

### 4.3. DNA Isolation

Tumor areas were marked by an experienced pathologist on an H&E-stained slide. Corresponding unstained tumor and paired normal tissues were macrodissected from formalin-fixed, paraffin-embedded (FFPE) 10 µm thick tissue sections. After overnight digestion with Proteinase K, DNA extraction was performed with the Maxwell 16 FFPE Plus Tissue LEV DNA Purification Kit (Promega, Mannheim, Germany) on the Maxwell 16 instrument (Promega) following manufacturer’s instructions as described before [37].

### 4.4. MMR Immunohistochemistry

For MMR- deficiency and proficiency testing 1 to 2 µm thick tumor sections were stained for *MLH1* (Clone: M1, Ventana), *MSH2* (G219-1129), *MSH6* (EPR3947) and *PMS2* (Clone: 44, Ventana) on a Ventana Benchmark stainer and were evaluated by an experienced pathologist. Samples were designated as dMMR if two out of 4 markers showed no staining and as pMMR if all four markers showed a distinct staining, respectively. Uncertain MMR status was equated by the loss of one marker, irregular staining of one or two markers or no availability of IHC.

### 4.5. MSI Testing by NGS

For NGS, DNA concentration was estimated by quantitative real-time PCR (qPCR) using the GoTaq qPCR Master Mix (Promega). Amplification of DNA was performed using the customized GeneRead DNAseq custom Panel V2 including microsatellite regions of the genes *BIRC3* (NR27), *STT3A* (NR22), *SLC7A8* (NR21), *MSH2* (BAT26) and *KIT* (BAT25) (GCGC-Panel) (Qiagen, Hilden, Germany), following the manufacturer’s instructions [38]. For library preparation, the Gene Read DNA Library I Core Kit and the Gene Read DNA I Amp Kit (Qiagen) were used. Libraries were ligated to NEXTflex DNA Barcodes (Bio Scientific, Austin, TX, USA) and sequenced on the MiSeq (Illumina, San Diego, CA, USA) with a MiSeq reagent kit V2 (300 cycles) (Illumina) following the manufacturer’s recommendations. MSI status was determined as described before for CRC with a threshold range indicating a factor of up to 2.6 as MSS and a factor above 3.0 as MSI-H [39]. Except sample 37, where no material was left after re-evaluation of the Idylla system, discordant results in cohort 2, showing MSS instead of MSI-H, were additionally tested with NGS.

### 4.6. MSI Testing by PCR and Fragment Length Analysis

#### 4.6.1. Bethesda

An in-house PCR protocol including primers for the mononucleotide markers BAT25 and BAT26, as well as the dinucleotide markers D5S346, D2S123 and D17S250, was performed with paired tumor and normal tissue DNA. For evaluation, PCR was followed by fragment length analysis on an ABI PRISM 3500 Genetic Analyzer and analyzed with the GeneMapper 4.1 analysis tool. (Applied Biosystems; Life Technologies, Darmstadt, Germany)

#### 4.6.2. MSI Analysis System, Version 1.2 (Promega)

The fluorescent PCR-based MSI Analysis System, Version 1.2 from Promega for co-amplification of five mononucleotide markers (BAT25, BAT26, NR21, NR24 and MONO27) and two pentanucleotide repeat markers (Penta C and Penta D) was investigated with paired tumor and normal tissue DNA following manufacturer’s instructions (Promega). PCR products were separated by capillary electrophoresis using the 3500 HID Genetic Analyzer (Applied Biosystems; Life Technologies Darmstadt, Germany). Data analysis was done using the GeneMapper^®^ Analysis Software (GeneMapper ^TM^ ID-X 1.6).

### 4.7. MSI Testing by Idylla

25–100 mm^2^ tumor areas of 10 µm tissue sections were used for the Idylla instrument (Biocartis) using the mononucleotide markers ACVR2A, BTBD7, DIDO1, MRE11, RYR3, SEC31A, SULF2 following the manufacturer’s instructions. The minimal amount of tissue sections and tumor cell content was defined according to manufacturer’s recommendations for CRC [40,41]. Reanalysis of selected tumor samples showing discrepancies with the testing approaches was done using 200 ng of extracted DNA. DNA amount was estimated by qPCR using the GoTaq qPCR Master Mix (Promega) and a final volume of 10–20 µL was loaded directly into the cartridges.

### 4.8. Concordance Level

For both cohorts the overall concordance was estimated in relation between PCR systems focusing on the final screening result, where at least two independent methods achieved a matching result.

### 4.9. Statistical Analysis

For better interpretation of NGS results, the average of deleted bases of 21 MSI-H EC samples of both cohorts were compared to the results of an unbiased set of 21 MSI-H CRC samples (Table 3). Statistical analysis was done using the GraphPad Prism v5 to calculate possible differences. A *p*-value of < 0.05 was considered as statistically significant according to the unpaired *t*-test.

## 5. Conclusions

In conclusion, this study showed that IHC is a valid first screen testing tool if the staining is obvious. All in all, using PCR testing systems established for CRC in EC is possible with careful estimation, validation and knowledge of limitations of each system. While the Bethesda and Promega panel user need to be aware of small peak alterations and small shifts events, laboratories using the Idylla or NGS should perform comprehensive validation before use.

## Figures and Tables

**Figure 1 cancers-13-01268-f001:**
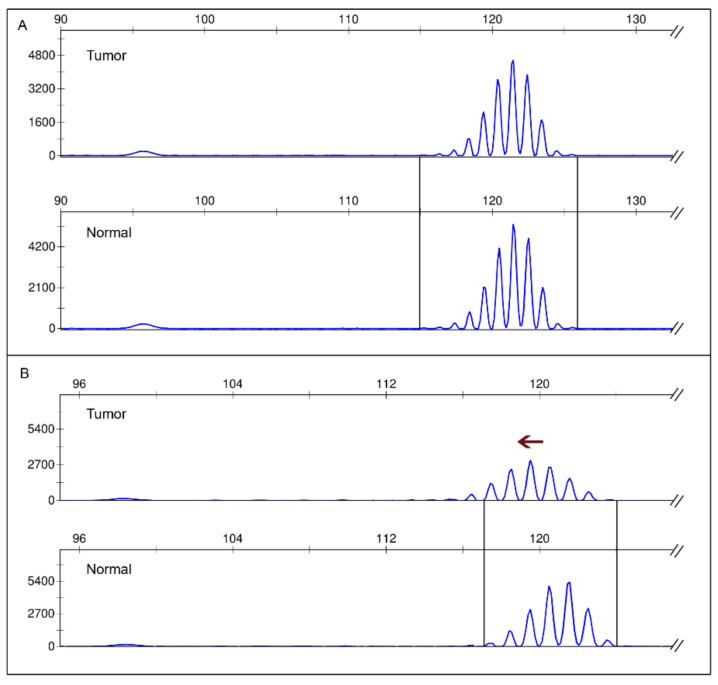
Doubtful microsatellite instability (MSI) profile of endometrial cancer (EC) after PCR and fragment length analysis using the Bethesda panel exemplarily shown for marker BAT25. (**A**) Stable BAT25 marker of the microsatellite stable (MSS) EC sample 11 (cohort 1) with its paired normal control. (**B**) Small left shift of the unstable BAT25 marker of the EC sample 1 (cohort 1) with its paired normal control. Arrow points to the small left shift indicating the marker as microsatellite instability-high (MSI-H).

**Figure 2 cancers-13-01268-f002:**
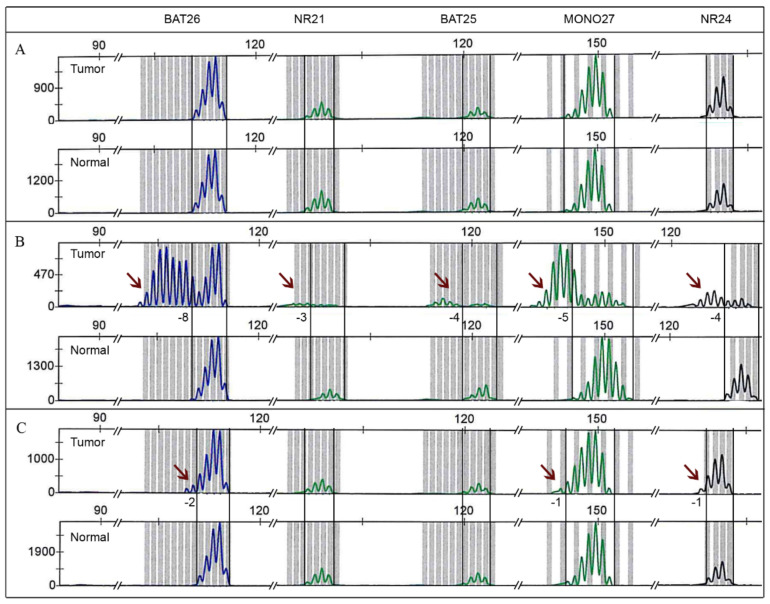
Different microsatellite instability (MSI) profiles of endometrial cancer (EC) after PCR and fragment length analysis using the Promega panel. All pictured tumor samples are compared directly with its paired normal tissue. (**A**) Sample 22 (cohort one) representing a microsatellite stable (MSS) profile. (**B**) Sample 4 (cohort one) showing a microsatellite instability-high (MSI-H) case with shifts in microsatellite repeat length comparable to colorectal cancer (CRC) samples. Available counted peaks are labeled at the bottom of each tumor sample. Arrows highlight each shift event in microsatellite repeat length. (**C**) Deviant MSI profile of sample 10 (cohort one) with small additional peaks of -1 to -2 nt.

**Figure 3 cancers-13-01268-f003:**
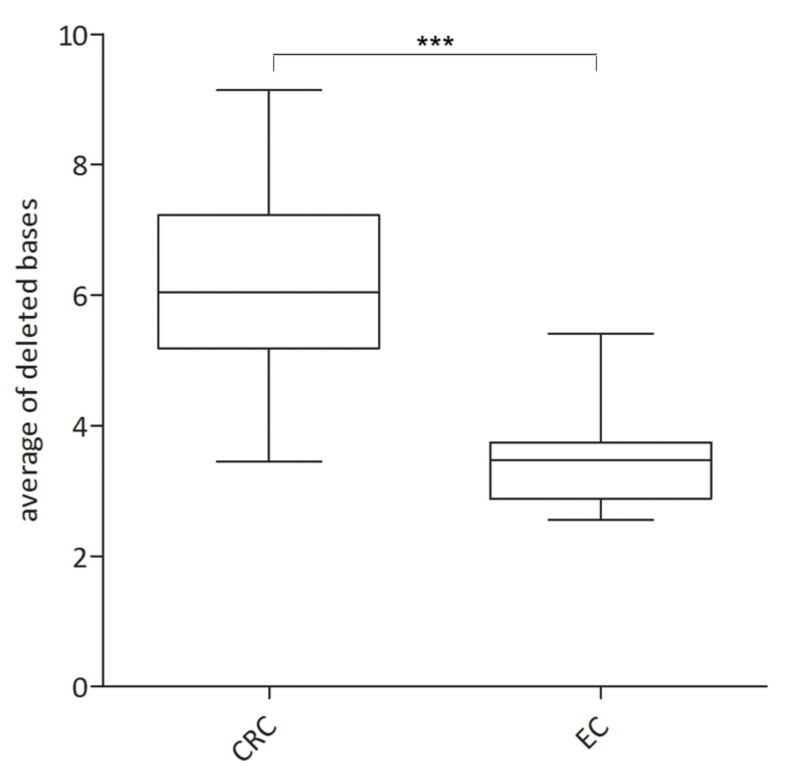
Statistical analysis of the average of deleted bases in colorectal cancer (CRC) and endometrial cancer (EC). Box plot graph was obtained via the GraphPad Prism software and *p*-value was calculated using the unpaired *t*-test with a significance level of *p* < 0.05. Statistical analysis revealed a significant divergence of the average of deleted bases between CRC and EC (*p*-value ≤ 0.0001). Level of significance is indicated by *******.

**Table 1 cancers-13-01268-t001:** Overview of mismatch repair (MMR) and microsatellite instability (MSI) status of cohort one after initial testing and re-analysis of four PCR-based testing methods including an in-house Bethesda panel, Promega panel, Idylla system and next generation sequencing (NGS) (customized GeneRead V2 panel). For re-analysis of the Idylla system, extracted DNA was used.

		Immunohistochemistry					Bethesda Panel		Promega Panel		Idylla		NGS		PCR Overall
Sample	TU	MLH1	PMS2	MSH2	MSH6	MMR Status	Initial Analysis	Re-Evaluation	Initial Analysis	Re-Evaluation	Initial Analysis	Re-Analysis	Initial Analysis	Factor	MSI Status
1	60	-	-	+	+	dMMR	MSI-H (4/5)		MSI-H (5/5)		MSI-H (6/7)		MSI-H (4/5)	4.1	MSI-H
2	80	-	-	+	+		MSS (5/5)	MSI-H (3/5)	MSI-H (5/5)		MSI-H (3/7)		MSI-H (2/5)	3.1	
3	25	-	-	-	-		MSI-H (3/5)		MSI-H (5/5)		MSI-H (2/7)		MSI-H (3/5)	3.4	
4	80	+	+	-	-		MSI-H (5/5)		MSI-H (5/5)		MSI-H (7/7)		MSI-H (5/5)	5.4	
5	50	-	-	+	+		MSI-H (2/5)		MSI-H (5/5)		MSI-H (5/7)		MSI-H (3/5)	4.4	
6	70	-	-	+	+		MSI-H (4/5)		MSI-H (5/5)		MSI-H (5/7)		MSI-H (3/5)	3.6	
7	90	-	+	-	-		MSI-H (4/5)		MSI-H (3/5)		MSI-H (4/7)		n.a.	n.a.	
8	80	-	-	+	+		MSI-H (4/5)		MSI-H (5/5)		n.a.	MSI-H (2/7)	MSI-H (4/5)	4.9	
9	/	-	-	+	+		MSI-H (5/5)		MSI-H (5/5)		n.a.	MSI-H (3/7)	MSI-H (3/5)	3.7	
10	80	-	-	+	+		MSI-H (2/5)		MSS (5/5)	MSI-H (3/5)	MSS (7/7)	MSI-H (3/7)	//	2.7	
11	70	-	-	+	+		MSI-H (5/5)		MSI-H (5/5)		MSI-H (7/7)		MSI-H (3/5)	3.8	
12	70	+	-	+	+	uncertain	MSI-H (4/5)	MSI-H (5/5)	MSI-H (4/5)		MSI-H (6/7)		MSI-H (3/5)	3.6	
13	60	*	*	+	+		MSI-H (3/5)		MSI-H (5/5)		MSI-H (6/7)		MSI-H (3/5)	3.1	
14	70	+	+	+	-		MSI-H (3/5)		MSI-H (5/5)		MSI-H (2/7)		MSI-H (3/5)	3.5	
15	40	/	/	/	/		MSI-H (3/5)		MSS (5/5)	MSI-H (4/5)	MSS (7/7)	MSI-H (3/7)	//	2.7	
16	40	/	/	/	/		MSI-H (3/5)		MSS (5/5)	MSI-H (3/5)	MSS (7/7)	MSI-H (4/7)	MSS (5/5)	2.6	
17	90	+	+	-	*		MSI-L (1/5)	MSI-H (2/5)	MSS (5/5)	MSI-H (2/5)	MSI-H (2/7)		//	2.8	
18	70	+	*	+	+		MSI-L (1/5)		MSS (5/5)	MSI-L (1/5)	MSI-L (1/7)		MSS (5/5)	2.6	MSI-L
19	20	+	-	+	+		MSS (5/5)		MSS (5/5)		MSS (7/7)		MSS (5/5)	2.3	MSS
20	40	+	-	+	+		MSS (5/5)		MSS (5/5)		MSS (7/7)		MSS (5/5)	2.3	
21	60	+	+	*	+		MSS (5/5)		MSS (5/5)		MSS (7/7)		MSS (5/5)	2.4	
22	30	+	+	*	*		MSS (5/5)		MSS (5/5)		MSS (7/7)		MSS (5/5)	2.2	
23	20	+	-	+	+		MSS (5/5)		MSS (5/5)		MSS (7/7)		MSS (5/5)	2.3	
24	70	n.a.	n.a.	n.a.	n.a.		MSS (5/5)		MSS (5/5)		MSS (7/7)		MSS (5/5)	2.2	
25	40	+	+	+	+	pMMR	MSS (5/5)		MSS (5/5)		MSS (7/7)		MSS (5/5)	2.3	

The next generation sequencing (NGS) factor represents the average of deleted bases of *BIRC3* (NR27), *STT3A* (NR22), *SLC7A8* (NR21), *MSH2* (BAT26) and *KIT* (BAT25). TU: Tumor cell content (+) intact staining of MLH1, PMS2, MSH2 or MSH6; (-) complete loss of expression; (/) no data available; (*) low staining intensity or partially loss; (//) threshold region; (n.a.) samples which were not analyzable. Microsatellite instability-high (MSI-H); microsatellite instability-low (MSI-L); microsatellite stable (MSS). Boxes highlighted in gray show the discrepancies between the different testing systems. Numerals within parentheses represent the number of stable and unstable markers in comparison to the absolute number of tested markers of each system, respectively.

**Table 2 cancers-13-01268-t002:** Average of the number of deleted bases in 21 colorectal cancers (CRCs) and endometrial cancers (ECs) after next generation sequencing (NGS).

Tumor	Sample	Cohort	Average of the Number of Deleted Bases (All)
CRC	1	none	3.99355
CRC	2	none	9.14104
CRC	3	none	5.078
CRC	4	none	8.09522
CRC	5	none	6.65126
CRC	6	none	7.4206
CRC	7	none	6.49183
CRC	8	none	6.73745
CRC	9	none	3.4492
CRC	10	none	5.59544
CRC	11	none	5.67546
CRC	12	none	3.51171
CRC	13	none	7.31834
CRC	14	none	6.02835
CRC	15	none	5.29999
CRC	16	none	4.90917
CRC	17	none	7.98234
CRC	18	none	7.13231
CRC	19	none	5.98217
CRC	20	none	6.04432
CRC	21	none	6.99662
EC	1	1	4.08461
EC	2	1	3.1351
EC	3	1	3.34857
EC	4	1	5.40648
EC	5	1	4.42547
EC	6	1	3.59148
EC	8	1	4.85327
EC	9	1	3.70891
EC	10	1	2.66275
EC	11	1	3.77505
EC	12	1	3.63687
EC	13	1	3.11685
EC	14	1	3.51233
EC	15	1	2.74571
EC	16	1	2.55675
EC	17	1	2.7671
EC	2	2	3.08447
EC	22	2	2.79111
EC	38	2	2.96433
EC	45	2	3.5554
EC	50	2	3.47716

**Table 3 cancers-13-01268-t003:** Overview of mismatch repair (MMR) and microsatellite instability (MSI) status of cohort two after initial testing including immunohistochemistry, an in-house Bethesda panel, the MSI status estimated by another institute and the MSI status using the Idylla system, as well as reanalysis of microsatellite instability-high (MSI-H) samples resulting in microsatellite stable (MSS) via the Idylla system.

		Immunohistochemistry					Bethesda Panel	MSI Status	Idylla		PCR Overall
								External			
Sample	TU	MLH1	PMS2	MSH2	MSH6	MMR Status	Initial Analysis	Initial Analysis	Initial Analysis	Re-Analysis	MSI Status
1	50	+	+	-	-	dMMR	/	MSI-H	MSI-H (6/7)		MSI-H
2	30	-	-	+	+		MSI-H (4/5)	/	MSS (7/7)	**; #	
3	50	-	-	+	+		MSI-H (4/5)	/	MSI-H (6/7)		
4	80	-	-	+	+		MSI-H (5/5)	/	MSI-H (6/7)		
5	70	-	-	+	+		/	MSI-H	MSI-H (5/7)		
6	70	-	-	+	+		MSI-H (3/5)	/	MSI-L (1/7)		
7	80	-	-	+	+		MSI-H (4/5)	/	MSI-H (4/7)		
8	90	-	-	+	+		MSI-H (5/5)	/	MSI-H (4/7)		
9	60	-	-	+	+		/	MSI-H	MSI-H (5/7)		
10	80	+	+	-	-		/	MSI-H	MSI-H (7/7)		
11	80	-	-	+	+		/	MSI-H	MSI-H (5/7)		
12	50	-	-	+	+		/	MSI-H	MSI-H (7/7)		
13	80	-	-	+	+		/	MSI-H	MSI-H (7/7)		
14	50	-	-	+	+		/	MSI-H	MSI-H (6/7)		
15	90	-	-	+	+		/	MSI-H	MSI-H (7/7)		
16	80	-	-	+	+		/	MSI-H	MSI-H (5/7)		
17	70	-	-	+	+		/	MSI-H	MSI-L (1/7)		
18	60	-	-	+	+		/	MSI-H	MSI-H (6/7)		
19	80	-	-	+	+		MSI-H (4/5)	/	MSI-L (1/7)		
20	70	-	-	+	+		/	MSI-H	MSI-H (4/7)		
21	80	-	-	+	+		/	MSI-H	MSI-H (2/7)		
22	50	-	-	+	+		MSI-H (4/5)	/	MSS (7/7)	MSI-L (1/7)	
23	50	-	-	+	+		MSS (5/5)	/	MSS (7/7)		MSS
24	80	+	+	-	-		MSI-H (4/5)	/	MSI-H (3/7)		MSI-H
25	60	-	-	+	+		MSI-H (4/5)	/	MSI-H (6/7)		
26	80	-	-	+	+		MSI-H (3/5)	/	MSI-H (4/7)		
27	70	-	-	+	+		MSI-H (4/5)	/	MSI-H (6/7)		
28	70	-	-	+	*		MSI-H (4/5)	/	MSI-H (2/7)		
29	80	-	-	+	+		MSI-H (4/5)	/	MSI-H (3/7)		
30	80	-	-	+	+		/	MSI-H	MSI-H (3/7)		
31	70	+	+	-	-		MSI-H (2/5)	/	MSI-H (2/7)		
32	80	-	-	+	+		MSI-H (3/5)	/	MSI-H (5/7)		
33	70	-	-	+	+		MSI-H (3/5)	/	MSI-H (2/7)		
34	90	+	+	-	-		MSI-H (5/5)	/	MSI-H (5/7)		
35	70	-	-	+	+		MSI-H (3/5)	/	MSI-H (4/7)		
36	60	-	-	+	+		MSI-H (5/5)	/	MSI-H (4/7)		
37	30	+	+	-	-		/	MSI-H	MSS (7/7)	MSS (7/7) **	
38	60	-	-	+	+		MSI-H (4/5)	/	MSS (7/7)	#	
39	70	-	-	+	+		/	MSI-H	MSI-H (4/7)		
40	80	-	-	+	+		/	MSI-H	MSI-H (4/7)		
41	80	-	-	+	+		MSI-H (5/5)	/	MSI-H (3/7)		
42	80	-	-	+	+		MSI-H (5/5)	/	MSI-H (5/7)		
43	80	/	/	/	/	uncertain	/	MSI-H	MSI-H (6/7)		
44	60	/	/	/	/		/	MSI-H	MSI-H (5/7)		
45	60	-	+	+	+		MSI-H (4/5)	/	MSS (7/7)	#	
46	50	/	/	/	/		/	MSI-H	MSI-H (5/7)		
47	80	/	/	/	/		/	MSI-H	MSI-H (5/7)		
48	80	+	+	+	-		/	MSI-H	MSI-H (3/7)		
49	40	-	+	+	+		/	MSI-H	MSI-H (2/7)		
50	80	n.a.	n.a.	n.a.	n.a.		MSI-H (5/5)	/	MSS (7/7)	#	
51	80	+	+	n.a.	n.a.		MSI-H (4/5)	/	MSI-H (6/7)		
52	80	*	*	+	+		MSI-H (2/5)	/	MSI-H (4/7)		
53	30	/	/	/	/		MSS (5/5)	/	MSS (7/7)		MSS
54	40	*	*	+	+		MSS (5/5)	/	MSS (7/7)		
55	70	+	-	+	+		MSS (5/5)	/	MSS (7/7)		
56	60	n.a.	n.a.	n.a.	n.a.		MSS (5/5)	/	MSS (7/7)		
57	50	*	*	*	*		MSS (5/5)	/	MSS (7/7)		
58	70	+	+	+	+	pMMR	/	/	MSS (7/7)		
59	80	+	+	+	+		MSS (5/5)	/	MSS (7/7)		
60	90	+	+	+	+		MSS (5/5)	/	MSS (7/7)		
61	60	+	+	+	+		MSS (5/5)	/	MSS (7/7)		
62	70	+	+	+	+		/	/	MSS (7/7)		
63	80	+	+	+	+		/	/	MSS (7/7)		
64	70	+	+	+	+		/	/	MSS (7/7)		
65	60	+	+	+	+		/	/	MSS (7/7)		
66	80	+	+	+	+		/	/	MSS (7/7)		
67	30	+	+	+	+		/	/	MSS (7/7)		
68	80	+	+	+	+		/	/	MSS (7/7)		
69	40	+	+	+	+		/	/	MSS (7/7)		
70	80	+	+	+	+		/	/	MSS (7/7)		
71	50	+	+	+	+		/	/	MSS (7/7)		
72	80	+	+	+	+		/	/	MSS (7/7)		
73	50	+	+	+	+		MSS (5/5)	/	MSS (7/7)		
74	40	+	+	+	+		/	/	MSS (7/7)		
75	80	+	+	+	+		/	/	MSS (7/7)		

TU: Tumor cell content; (+) intact staining of MLH1, PMS2, MSH2 or MSH6; (-) complete loss of expression; (/) no data available; (*) low staining intensity or partially loss; (n.a.) samples which were not analyzable; (**) 30% tumor cell content; (**#**) samples without adequate DNA concentration after q-PCR measurement. Microsatellite instability-low (MSI-L). Boxes highlighted in gray show the divergence between the Idylla system and MSI status estimated with the in-house Bethesda panel or another institute. Numerals within parentheses represent the number of stable and unstable markers in comparison to the absolute number of tested markers of each system, respectively.

## Data Availability

Not applicable.

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
