# Peer review of "Managing Difficulties of Microsatellite Instability Testing in Endometrial Cancer-Limitations and Advantages of Four Different PCR-Based Approaches"

_cancers, 2021, doi:10.3390/cancers13061268_

Round 1

Reviewer 1 Report

The manuscript has been edited according with reviewers' suggestions.

Reviewer 2 Report

The manuscript “Managing difficulties of microsatellite instability testing in endometrial cancer- limitations and advantages of four different PCR based approaches (cancers- 1062391)” aims at comparing four different techniques (Bethesda, Promega, Idylla and NGS) for microsatellite instability assessment in endometrial cancer in order to provide reliable tools that guide medical decisions in those samples with uncertain immunochemistry classification. Authors analyze two cohorts of 25 and 75 samples, respectively, comparing the four different molecular techniques in cohort one and validating the Idylla system in cohort 2.

The points covered in the manuscript are of interest in current research in endometrial cancer. The paper is well written, English is appropriate and the questions addressed in the paper can improve clinical management of endometrial cancer patients.

I have carefully revised the responses provided by the authors, especially those modifications concerning the structure of the paper to ease its reading (relocation of tables and figures, rewriting of material and methods section…) and explanations on the aim of the paper, study design, samples included in each comparison, inclusion criteria for each comparison…

After carefully reviewing all these points modified by authors (which cover the majority of questions pointed out in the per-review process), I would recommend the paper for publication in “Cancers” journal. 

Reviewer 3 Report

I think comments of reviewer 2 have been addressed satisfactorily and the manuscript can be accepted.

This manuscript is a resubmission of an earlier submission. The following is a list of the peer review reports and author responses from that submission.

Round 1

Reviewer 1 Report

The authors wanted to verify the applicability in routine diagnostics of four different PCR-based methods of analysis of the Microsatellite instability of endometrial cancer.

The scientific content of this manuscript is significant for researchers in the field of diagnosis and treatment of endometrial cancer.

The manuscript is original and pragmatic at the same time and the quality of the presentation is high.

A few points:

  • The manuscript needs English language editing for minor issues.
  • At the end of the summary the Authors should explain how the findings
    from this research may impact the research community.
  • References should be formatted according to “MDPI Reference List and
    Citations Style Guide”.
  • On page 12 of 20 line 348: typographical error “4.4. MSI testing by
    NGS”
  • remove underscore space before MSI
  • On page 12 of 20 line 376: typographical error “4.6. MSI Testing by
    Idylla” 
  • remove first line indent and precede an empty line.

In my opinion the manuscript needs only minor text editing.

Reviewer 2 Report

Dear editor,

The manuscript “Managing difficulties of microsatellite instability testing in endometrial cancer- limitations and advantages of four different PCR based approaches (cancers- 1062391)” aims at comparing four different techniques (Bethesda, Promega, Idylla and NGS) for microsatellite instability assessment in endometrial cancer in order to provide reliable tools that guide medical decisions in those samples with uncertain immunochemistry classification. Authors analyse  two cohorts of 25 and 75 samples, respectively, comparing the four different molecular techniques in cohort one and validating the Idylla system in cohort 2.

The points covered in the manuscript are of interest in current research in endometrial cancer. The paper is well written, English is appropriate and the questions addressed in the paper can improve clinical management of endometrial cancer patients.

Nevertheless, I would recommend the paper for publication in “cancers” only after some points are taken into consideration and solved, what would be a major revision:

Abstract

I would suggest specifying that the seven-marker system corresponds to the Idylla system

Introduction

Regarding the aim of the study, it is not clear which is the aim of cohort 2, since the authors state what they already achieved, instead of their goal.

Additionally, the aim of the study is confusing as it is stated. When reading the “Study design” section, it seems that the study aims to assess the performance of the Idylla system and NGS in cohort 1 (considering as the reference IHC, Bethesda or Promega Panel) and to validate the Idylla panel in cohort 2. This is more evident when authors mention that “discordant results of the Idylla system were re-evaluated”. Please reconsider what the true aim of the study is.

On the other hand, there is no objective regarding the NGS of CRC and EC tumours to optimize the threshold range. Please, reconsider including this  in the aims section.

Results
On my experience, “study design” section corresponds to the material and methods section. I do understand that authors are providing results of the MMR classification obtained in each cohort. Therefore, authors might reconsider the heading of this section or adding a “study design section” in material and methods with the sentences merely corresponding to the design of the experiment and leaving the results achieved in this renamed section.

I could not help but observe the different proportion of MMRd, MMRp and uncertain samples in cohort 1 (44%, 4% and 52%) and in cohort 2 (56%, 24% and 15%). Authors might explain the inclusion criteria for samples in cohort 1 and in cohort 2 to avoid any possible source of bias.

Regarding study design and to minimize the sources of bias, authors are to explain whether or not re-evaluation of the different panels in cohort 1 was blinded with respect to the results of the other panels to which a particular panel is compared. Please, clarify in the materials and methods section of the manuscript.

I would recommend adding a table with the clinic-demographical characteristics of patients of study (including the 20 CRC patients).

The fact that Table 1 and Table 2 are plotted on a row makes it difficult to follow the results section. I would suggest moving table 1 wherever within section 2.3 and table 2 wherever within section 2.4.

Once again, section 2.2. seems to belong to Material and Methods section. No results of concordance are reported. I would suggest adding an additional column into tables 1 and 2 depicting the MSI status agreed among the different molecular methods (as it is depicted for IHC results). This might ease the reading of the manuscript.

Regarding table 1 and table 2, I have not been able to find an explanation for fractions in parentheses. Please describe at table footnote.

Table 1, sample 12. Why MSI-H (4/5) for the Bethesda testing is highlighted in Grey? It does not seem to show any discrepancy since the rest of diagnoses in the line are also MSI-H. Please reconsider.

Line 170, section 2.3.2. Please, clarify the “two additional testing methods” Bethesda panel is compared to. Accordingly, clarify (line 175) which other systems is the Promega panel compared to.

Which were the inclusion criteria of the 20 CRC samples and 20 EC samples for NGS?  Please, reconsider moving table 3 and figure 2 to supplementary material, since it does not fit into the main goal of the manuscript and distracts the reader from it. Adding to my former comment on the inclusion criteria for cohort 1 and 2, I am surprised to discover that 5 samples analysed by NGS belong to cohort 2, in which no NGS analyses had been described. Why are not all the sequenced samples from cohort 1 included in table 3? (I can count at least 21 with NGS factor values) How many samples from cohort 2 have been sequenced? Are all they included in table 3 and in the statistical analyses depicted in figure 2? Finally, I would recommend ordering the samples in table 3 by sample ID, as occurred for the CRC cohort. 

Line 220 and 221. Reconsider the accuracy of the samples mentioned in this sentence.

Line 235. Consider changing wasn’t to was not.

Discussion

Line 253. I assume that authors refer to the Idylla system. Please clarify.

Figure 3 introduce new results, which are not to be in the discussion section. Please reconsider moving this figure to the results section.

Material and methods

The Promega panel employed has to be specified (as it is in line 114-115)
